# Identification of Epileptic EEG Signals Using Convolutional Neural Networks

**Rahib Abiyev [1,\*], Murat Arslan [1], John Bush Idoko [1], Boran Sekeroglu [2]**  **and Ahmet Ilhan [1]**

[1]   Applied Artificial Intelligence Research Centre, Department of Computer Engineering, Near East University, North Cyprus, 10 Mersin, Turkey; murat.arslan@neu.edu.tr (M.A.); john.bush@neu.edu.tr (J.B.I.); ahmet.ilhan@neu.edu.tr (A.I.)

[2]   Department of Information Systems Engineering, Near East University, North Cyprus, 10 Mersin, Turkey; boran.sekeroglu@neu.edu.tr

\*   Correspondence: rahib.abiyev@neu.edu.tr; Tel.: +90-39-2223-6464

**Abstract:** Epilepsy is one of the chronic neurological disorders that is characterized by a sudden burst of excess electricity in the brain. This abnormality appears as a seizure, the detection of which is an important research topic. An important tool used to study brain activity features, neurological disorders and particularly epileptic seizures, is known as electroencephalography (EEG). The visual inspection of epileptic abnormalities in EEG signals by neurologists is time-consuming. Different scientific approaches have been used to accurately detect epileptic seizures from EEG signals, and most of those approaches have obtained good performance. In this study, deep learning based on convolutional neural networks (CNN) was considered to increase the performance of the identification system of epileptic seizures. We applied a cross-validation technique in the design phase of the system. For efficiency, comparative results between other machine-learning approaches and deep CNNs have been obtained. The experiments were performed using standard datasets. The results obtained indicate the efficiency of using CNN in the detection of epilepsy.

**Keywords:** epilepsy; deep learning; convolutional neural networks; electroencephalography

## 1. Introduction

Epilepsy is a neurological disorder that is capable of causing spontaneous seizures [1]. Based on information given by the World Health Organization (WHO), over 50 million individuals have suffered an epileptic attack [2]. Patients with epilepsy often suffer from loss of consciousness and convulsive movements that can lead to serious physical injury. Early identification of this disease is a crucial health problem. The capability of detecting the occurrence of seizures can improve therapeutic treatment possibility and also the quality of life of epileptic patients. Seizure detection ability allows systems of intercession [3,4] to be executed for a set of patients for whom surgery or medication have zero impact. Currently, 70% of individuals suffering from epilepsy are able to manage the disease with the right medicine [3]. Pre-surgical evaluations are usually performed to ascertain the location of the disorder. This is done by utilizing a combination of physical exam, history, electroencephalogram (EEG), neuroimaging and other techniques [4,5]. These studies are insufficient in some patients; hence, focusing on the brain regions susceptible to epilepsy (ictal activity) through intracranial EEG is usually recommended. A longer period of hospitalization is regularly needed to detect enough seizures by utilizing intracranial electrodes. If an epileptic seizure is detected, surgical excision is recommended to the patient. In clear-cut lesion scenarios seen on neuro-imaging, surgery provided about 80% of treatment for epileptic patients [6].

A commonly used test for the identification of epilepsy is the use of EEG signals that provide details of the brain's electrical activity [1,7]. EEG signals are appropriate for describing the state of the brain and can be utilized to examine the brain function. Using the EEG signals, doctors can identify seizure signals and also inter-ictal (between seizure) and ictal (during a seizure) abnormalities in epileptiform. Sometimes, the excess electrical discharge causes the interruption of some brain functions. For this reason, timely and accurate identification of epilepsy at the early stage is necessary for such patients. It allows a reduction in the risk of complication of seizure-related problems. Epileptic seizure detection using EEG signals allows the identification of abnormalities in neurons and evaluates the physiological state of the brain of a human being. Traditionally, the identification of EEG signals is carried out by the visual interpretation of experts (doctors). However, this procedure is time-consuming and can sometimes be subject to human errors. Hence, automation of the identification process of an epileptic seizure using EEG signals becomes an important problem. Here the problem basically involves the extraction of distinguished features of EEG signals for seizure detection [7–9]. The accuracy of the designed model is determined by the detection of the EEG signal features [1].

Recently, more enhanced techniques indicating high sensitivities of seizure detection with the application of statistical validation have been developed. Statistical evaluation is an ongoing prediction technique requirement inspired by Mormann [1]. The classification of EEG signals using a statistical framework and seizure prediction algorithms has been presented in [1]. These approaches have led to the establishment of the generally acknowledged statistical system [10,11]. The reference [11] presented a random prediction model of seizure and gave comparative results of different models.

A series of research studies on seizure identification have been conducted using various signal processing and machine-learning techniques. In [12], the authors decomposed electroencephalogram signal into its frequencies using wavelet transform (WT), and extracted correlation dimension, standard deviation, and largest Lyapunov exponent, which are the three distinctive features from the signals. Different techniques are applied for the classification of epileptic seizures and the wavelet-chaos-neural network approach has depicted good performance. In [13], fast Fourier transform (FFT) was utilized for feature extraction and decision tree classifier was utilized to classify these features. The paper [14] applied two distinctive approaches: principal component analysis (PCA) and genetic algorithm (GA) on several linear and non-linear algorithms. Application of principal component analysis on a non-linear algorithm produced better outcomes compared to a genetic algorithm. Reference [15] depicted a review of wavelet-based approaches to detect epileptic seizures. Reference [16] proposed an approach that applies training and testing sets to classify and detect electroencephalogram non-seizure and seizure signals via the extraction of higher-order spectral features. The support vector machine (SVM) was applied for classification purposes [16]. Reference [17] used empirical mode decomposition to analyze EEG signals. The Hilbert–Huang transform (HHT) was used for the extraction of intrinsic mode functions. These functions were utilized as features to differentiate EEG signals. Reference [18] compared brain activities using distinctive brain regions and analyzed groups of ECG time series. In [19], using the non-linear dynamics of EEG signals quantified by correlation coefficients and largest Lyapunov exponent and wavelet-based methodology, the analysis of EEG signals was performed. Reference [20] presented logistic model trees that use statistical features based on an optimal allocation technique for detecting seizure from EEG signals. Some statistical features were extracted from EEG signals and these features were utilized as input in a logistic model tree (LMT) for epileptic seizure identification. The presented method was tested using benchmark EEG dataset. The papers mentioned above depict different methodologies used for extraction of features and classification purposes. The accuracies of these designed models are important performance characteristics of the designed systems. In the paper [21] that used discrete wavelet transform (DWT), the EEG signals were fragmented into frequency sub-bands leading to the extraction of statistical features. The PCA, independent components analysis (ICA) and linear discriminant analysis (LDA) were utilized for data size reduction. Using the obtained features, SVM was applied to classify epileptic seizures from non-epileptic seizures. In [22], data points of Universum were created via the selection of Universum

from a dataset of EEG itself otherwise known as inter-ictal EEG signals. Using feature extraction techniques and Universum SVM, the authors in [22] performed seizure classification. In [23], a group of important features was extracted from the EEG data, utilizing NCA (neighbourhood component analysis). The system performance was assessed utilizing SVM, AdaBoost (adaptive boosting), K-NN and random forest classifiers. References [24,25] used convolutional neural networks (CNNs) to analyze the EEG signals. The CNN model was utilized to extract features and the extracted features were used to perform the classification of normal, preictal and seizure classes. The papers [26,27] used a machine learning approach for the detection of real-time seizure from intracranial EEG signal. Here, the authors extracted spectral and temporal features and utilized them for the training of the pattern recognition component. Reference [28] used linear discriminant analysis and triadic wavelet decomposition-based features, and K-NN classifiers to classify the signals of the seizure.

The major objective of this research is to detect seizures caused by epilepsy. The seizure detection ability allows systems of intercession to be executed for a set of patients where surgery or medication have had zero impact. The structures of the above-constructed systems are complicated and basically, they include the extraction of features and classification stages. In this study, these two methodologies are combined in the body of CNN to identify seizure from EEG signals.

CNN is a machine-learning method that is based on learning representation where the framework naturally learns and finds the features required for detection from the numerous layers processing input datasets [29]. Deep learning has officially demonstrated its capacity and has outperformed human reasoning in image and audio recognition problems [29,30]. It has been utilized in numerous complex applications of machine learning, such as the diagnosis of Alzheimer's at an early stage [31], detection of chest diseases [32], concrete comprehensive strength estimation [33]. Additionally, numerous expansive technological companies, for example, Apple (Cupertino, CA, USA), Google (Mountain View, CA, USA), Baidu (Beijing, China), IBM (Armonk, NY, USA), Microsoft (Redmond, WA, USA), Netflix (Los Gatos, CA, USA) and Facebook (Menlo Park, CA, USA) have grasped and used deep learning in their studies [34–36].

The use of artificial intelligence elements in the development of computer models for the identification of diseases has attracted the interest of numerous researchers [37–42]. During the design phase of these retrospective models, functional connectivity derived from EEG signal recordings and dynamical simulations was combined. The goal of our research is to predict surgical outcomes by utilizing a convolutional neural network learning capability to detect epileptic seizure.

Recently, a number of papers have been published for the identification of epileptic seizures [43–47] using machine-learning techniques. Different approaches including GAs [43], Fourier–Bassel series expansion of EEG signals [44], particle swarm optimisation (PSO)-based neural networks [45], Hilbert transform and vector quantization based classifier [46] and wavelet filters [47] have been applied to solve epileptic seizure identification problems and have recorded high accuracy. The basic contributions of our study include but are not limited to the following: we present a productive technique based on convolutional neural network for the pre-processing of a raw EEG medically related dataset and improve the performance of the system. Here, we formulate a rule to enable the CNNs to work well in the detection of epileptic seizures with least feature control. We also make available a framework that functions admirably on different domains. Another contribution of our paper is that it reveals factors that determine accurate detection of a seizure in other related datasets. Furthermore, we utilize the framework as a network measure to propose another resection method to diagnose epilepsy seizures.

The remaining part of the paper is organised as follows. Section two demonstrates CNNs utilized for epileptic EEG signals' identification. Section three presents the simulation results. Finally, section four summarizes the entire content.

## 2. Convolutional Neural Networks (CNNs)

A CNN is a multilayer neural network (NN) architecture which incorporates one or more convolution, max-pooling and fully-connected layers. The convolution layers represent the core

building block of the network and have a hierarchical structure. In this paper, a one-dimensional CNN is used for the detection of epilepsy. Figure 1 depicts the structure of the one-dimensional double CNN. The input of the first convolution layer represents the input space and the output represents the feature map. The input and output of the subsequent convolutional layers are feature maps of input space. In Figure 1, we explore the four convolutional layers. The complex features of input space are represented by using the stacked hierarchical structure of convolutional layers. Features obtained from the convolutional layers are fed to the pooling layers. A rectified linear unit (ReLU) activation function is applied to the obtained feature map. In this layer, the relevant features are retained and the rest are discarded. The obtained features are transformed into a one-dimensional array known as a feature vector. The feature vector is a one-dimensional array and is the input for the fully connected network. This layer calculates the output of the CNN [48,49].

Considering the formulation of the above operations, and using the input signal and local kernels, the output of the convolution layer will be computed. Each output is calculated by the dot product of input and kernel (filter) weight coefficients. Depending on the number of kernels, the volume will be [*p,q*]. Here, *q* is a number of kernels. The feature maps are determined by using kernels (filters). For $x_{i,j}^{l}$ input space, the feature value $z_{i,j,k}^{l+1}$ at the location (*i, j*) in the *k*th feature map of the *l* + 1th convolutional layer is calculated by:

$$z_{i,j,k}^{l+1} = w_{k}^{lT} \cdot z_{i,j}^{l} + b_{k}^{l} \tag{1}$$

where weight $w_{k}^{l}$ is the vector and $b_{k}^{l}$ is the bias term of the *k*th filter of the *l*th layer. In the first layer, *l* = 1 and $z_{i,j}^{l} = x_{i,j}^{l}$. The kernel generates the feature map $z_{i,j,k}^{l+1}$ as shown in [50–52]. The second convolutional operation is applied to the output of the first layer. Unlike multi-layer networks, the activation function is used for non-linear transformation of signals of CNN to detect non-linear features. For this aim, the non-linear activation function is applied for the transformation of (1).

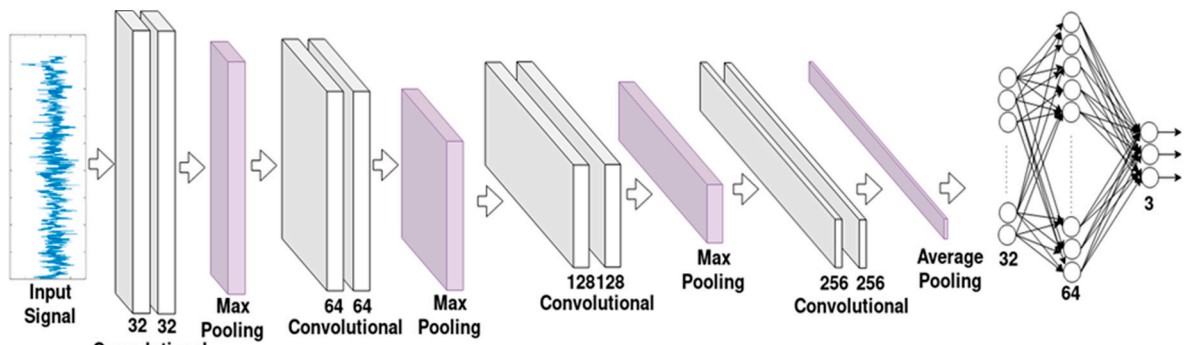

**Figure 1.** One-dimensional convolutional neural network (CNN).

$$a_{i,j,k}^{l+1} = a\left(z_{i,j,k}^{l+1}\right) \tag{2}$$

where $z_{i,j,k}^{l+1}$ is calculated by the formula (1). Typical activation functions are tanh, sigmoid and ReLU [52]. In the paper, the non-saturated activation function called ReLU is utilized. It is denoted by $max\left(0, z_{i,j,k}^{l+1}\right)$. ReLU prunes the negative values to zero and holds positive values. ReLU activation function produces quicker results than the sigmoid or tanh activation functions and speeds up execution time [50–52]. This function can induce the sparsity in the hidden units and obtain sparse representations of the network. The use of ReLU allows easy training of the deep networks.

By reducing the feature map resolution, the pooling layer achieves shift-invariance. A pooling layer is often used between the convolutional layers. The feature map of a convolutional layer is

converted to the feature map of a pooling layer. For each feature map $a^l_{i,j,k}$ the pooling layer output is determined by:

$$y^{l+1}_{i,j,k} = pool\left(a^{l+1}_{m,n,k}\right), \forall (m,n) \in R_{i,j} \tag{3}$$

where $R_{i,j}$ is a local neighbourhood around the location $(i,j)$. In the literature, average pooling and max pooling are often used as typical pooling operations. In this paper, max pooling is used. As a result of using the above operations, the feature map will be obtained. We apply convolutional, ReLU and pooling operations repeatedly. In the first convolutional layer, the kernels are used to determine low-level features (edges), but the kernels in the subsequent layers are designed to determine more abstract features. If many convolutions, ReLU and pooling layers are used, more detailed features are determined.

The last layer of CNN is a classifier known as the dense layer. This layer is called a fully connected layer which performs the classification of features. Individual features obtained from the convolutional layers are applied as an input to the fully connected layer. For this purpose, the output of the convolutional part of the CNN is converted into a one-dimensional feature vector and used for the classification. Conversion of features to a one-dimensional (1D) vector is called flattening. In the fully connected layer, all neurons of the previous layer are connected to the neurons of the current layer. At this time, all the neurons of the current layer are connected to the neurons of the next layer. Softmax function is used for classification in the fully-connected network. The output of a fully connected network will be clustering results.

$$y^{(l)}_i = f(z^{(l)}_i) \text{ with } z^{(l)}_i = \sum_{i=1}^{m^{(l-1)}_i} w^{(l)}_{i,j} y^{(l-1)}_i \tag{4}$$

where $w^{(l)}_{i,j}$ are the weight coefficients between the neurons of the fully connected layers, $f$ is a transfer function representing the non-linearity.

Training of the CNN starts after determining the output signals. During training, the loss function is determined in the output of the CNN. This loss function is propagated back so as to change the network's interconnections. Training is performed using the RMSprop learning algorithm presented in [53]. This technique is an extension of the stochastic gradient technique attempting to solve the rapid decreasing (vanishing) learning rate issue. The algorithm solves the problem by normalizing the gradient itself and using the average of the squared gradient. The update of the network's interconnections is based on the reduction of the loss function.

$$\theta_j = \theta_j - \epsilon \frac{\nabla_\theta L(\theta)}{\sqrt{g_{t+1}} + 1e^{-5}}; \quad g_{t+1} = \alpha \cdot g_t + (1 - \alpha)\nabla_\theta L(\theta)^2 \tag{5}$$

where $\theta_j$ are network parameters, $L(\theta) = \frac{1}{N}\sum_{i=1}^N l\left(\theta; yd^{(i)}, y^{(i)}\right)$ is the loss function calculated at the output of the network, $\alpha$ is the decay rate, $\epsilon$ is the learning rate, $yd$ and $y$ are the desired and current values of the network output, and $N$ is the number of training pairs. During training, CNN parameters are determined.

## 3. Simulation

The CNN described above is utilized for epileptic seizure detection. We explored the Bonn University (BU) EEG database [18] and the CHB-MIT scalp EEG database [26] in this research. The first data were downloaded from the BU EEG database [18]. The BU data were obtained from five patients and each dataset comprises 100 EEG signals. Every signal is captured within 23.6 s. In the simulations, EEG signals are generated using the same 128 channel amplifier. The data is digitized at a resolution of 12 bits and of 173.6 samples per second. The acquisition system has a bandwidth of 0.5 Hz to 85 Hz. Every dataset has 4096 (23.5 × 173.6) sampling points for 23.5 s. The total number of EEG

signals used in the simulation is 300; 100 normal signals (Set B), 100 preictal (Set D) and 100 seizures (Set E). After visual inspections, the signals are obtained from multichannel encephalogram recordings. The normal data set includes EEG signals obtained from five healthy instances, each of which contains 100 cases. The preictal file consists of 100 data obtained from five individuals with epilepsy who did not have epilepsy at the time of measurement. Similarly, the seizure class comprised 100 instances with the same subjects who had epilepsy at the time of data acquisition. Figure 2a–c depicts seizure, preictal and normal samples of EEG signals. As shown, EEG signals are high order non-linear and non-stationary signals that have a very complex structure. For the classification of such signals, the researchers try to extract basic distinguishing features and use these features for classification purposes. Analysis of the signals can sometimes be difficult. These processes are tedious and time-consuming. Hence, the accurate and fast identification of these signals is important. In this paper, a CNN is developed for the classification of EEG signals.

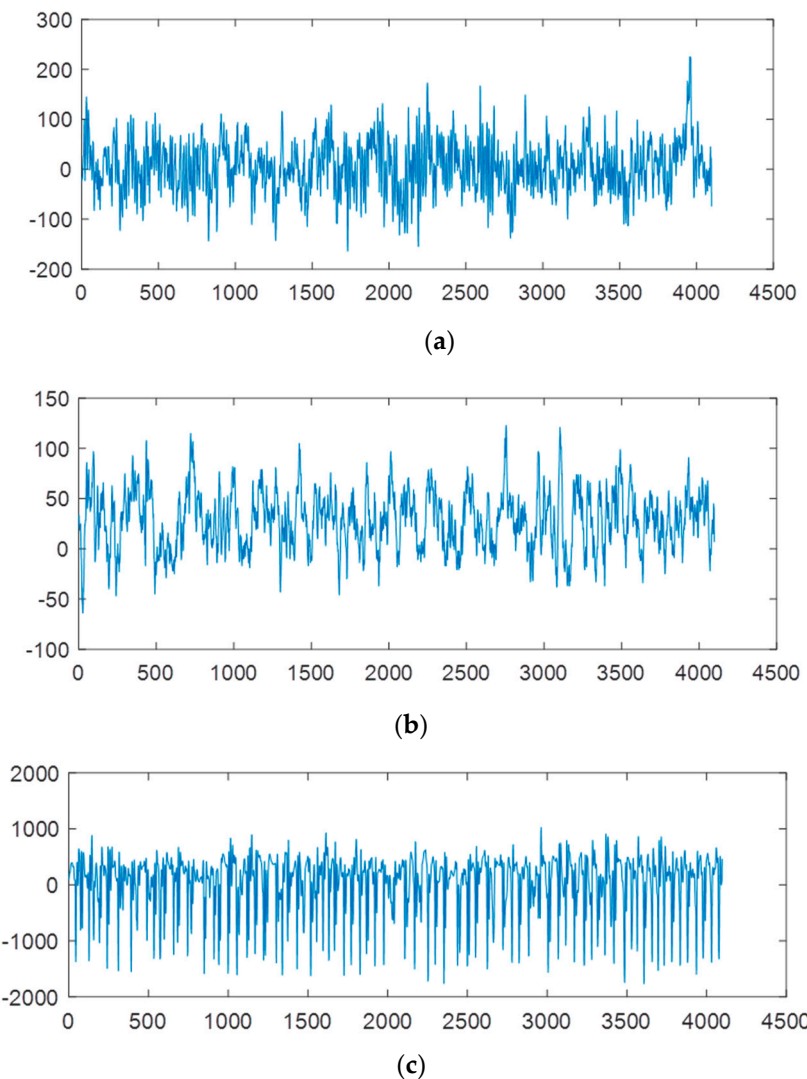

**Figure 2.** Normal (**a**), preictal (**b**) and seizure (**c**) signals.

The structure of the CNN used for the detection of seizures is presented in Table 1. The CNN includes four convolutional layers, global average pooling and the fully connected network. For CNN training, the dataset was divided into two: 90% and 10%. We utilized 90% of the available data for training while 10% of the available dataset was utilized for testing. From the 90% data assigned for

training, 70% was used for training and 30% for validation. Formulas (1)–(4) were applied to find the output signals of the CNN.

Using the CNN structure, RMSprop (5) algorithm and EEG signal, the learning of the weight parameters of CNN was performed. Keras, a powerful deep-learning library that runs on top of TensorFlow, was utilized for modelling. During training, the batch size for the data was selected as 100, and this was utilized for each of the training updates.

Since the input data is a one-dimensional brain signal, the CNN model is designed to accept one-dimensional data of size 4096. Z-score normalization was applied for the scaling of each input signal and the enhancement of model generalization was realized. After obtaining the input data, the first convolutional operation was applied to the input data and then the second convolutional operation was applied to the result of the first convolutional operation. Afterwards, the pooling was applied to the outputs of the convolutional layer. Pooling decreases the dimensions of the data. Increasing the number of convolutional layers will allow us to obtain more deep features, but this also increases the computational time. The fully connected network with three layers is applied for classification purposes.

**Table 1.** Description of the CNN structure.

| Layer (Type) | Output Shape | Parameters |
|---|---|---|
| conv1d_73 (Conv1D) | (4095, 32) | 128 |
| conv1d_74 (Conv1D) | (4093, 32) | 3104 |
| max_pooling1d_28 | (MaxPooling (1364, 32)) | 0 |
| conv1d_75 (Conv1D) | (1362, 64) | 6208 |
| conv1d_76 (Conv1D) | (1360, 64) | 12,352 |
| max_pooling1d_29 | (MaxPooling (453, 64)) | 0 |
| conv1d_77 (Conv1D) | (451, 128) | 24,704 |
| conv1d_78 (Conv1D) | (449, 128) | 49,280 |
| max_pooling1d_30 | (MaxPooling (149, 128)) | 0 |
| conv1d_79 (Conv1D) | (147, 256) | 98,560 |
| conv1d_80 (Conv1D) | (145, 256) | 196,864 |
| global_average_pooling1d_10 | (256) | 0 |
| dropout_1(Dropout) | (256) | 0 |
| dense_28 (Dense) | (32) | 8224 |
| dense_29 (Dense) | (64) | 2112 |
| dense_30 (Dense) | (3) | 195 |

The CNN was trained using 150 epochs. Figure 3 depicts the simulation results of loss function and accuracy. Table 2 depicts the simulation results of the CNN. During training, the value obtained for the loss function was $3.8277 \times 10^{-11}$. The value of the loss function obtained for validation data was 0.0237, and that for the test data was 0.013878. For the test data, the accuracy was 96.67%, specificity was 98.33%, and sensitivity was 96.67%.

For comparative analysis, we used the same data to train and test the SVM and NN models. We used the linear SVM with the linear kernel function. We also applied a three-layer NN structure with 18 hidden neurons and a Gaussian activation function. Several experiments were performed using the settings of the previous experiments. Table 3 includes comparative results of different models using the epilepsy data sets. As shown in the table, the value of the loss function for CNN was 0.013878 less than that of SVM, which was 0.25. Additionally, for NN, the loss function obtained was 0.1828. The accuracy result for SVM and NN was 75%, but that of CNN was 96.67%. The results obtained indicate the efficiency of using CNN in the identification of seizures.

In the second simulation, the design was also performed utilizing cross-validation. The structure of CNN used in the simulation was the same as given in Table 1. Ten-fold cross-validation was utilized during simulation. The EEG data was split into 10 equal proportions. Nine out of 10 were utilized

for training purposes and the remaining one portion was used for testing. The RMSprop learning algorithm was applied for training.

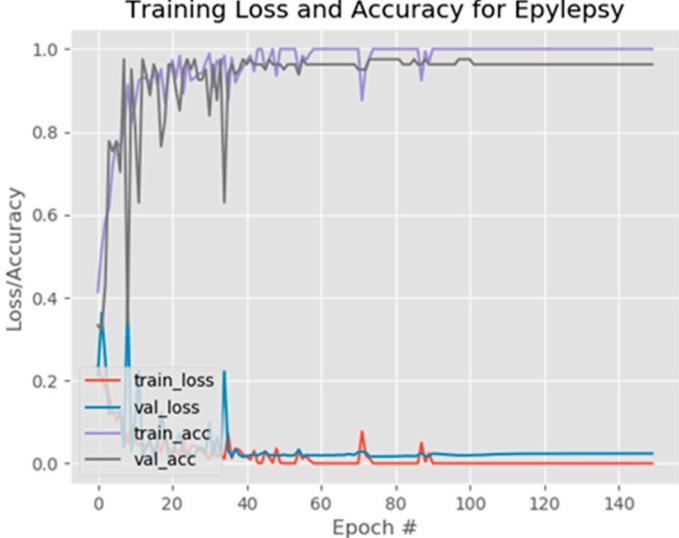

**Figure 3.** Simulation results of loss function and accuracy.

**Table 2.** Simulation results.

|  | Loss Function | Sensitivity (%) | Specificity (%) | Accuracy (%) |
|---|---|---|---|---|
| Train | $3.8277 \times 10^{-11}$ | 100 | 100 | 100 |
| Validation | 0.0237 | 96.30 | 98.15 | 96.30 |
| Testing | 0.013878 | 96.67 | 98.33 | 96.67 |

**Table 3.** Comparative results of different models.

|  | Loss Function | Sensitivity (%) | Specificity (%) | Accuracy (%) |
|---|---|---|---|---|
| SVM | 0.25 | 79 | 75 | 75 |
| NN | 0.1828 | 80 | 70 | 75 |
| CNN | 0.013878 | 96.67 | 98.33 | 96.67 |

Abbreviation: SVM: support vector machine; NN: neural network.

The demonstrated accuracy, sensitivity and specificity values are averaged over ten simulations. Figure 4 depicts the value of loss function (curves that are shown at the bottom of the figure) and the value of accuracy (curves that are shown at the top of the figure) obtained from the training and validation data. Table 4 contains the results of the simulations of CNN using the cross-validation approach.

The simulation was repeated 10 times. For the test data, the average accuracy rate is 98.67%, sensitivity is 97.67% and specificity is 98.83%. For comparative analysis, our experimental results were compared with the results of the same seizure classification problem performed by other authors. Table 5 includes the performances of different frameworks used for the detection of seizures. As shown in the table, the proposed model has good characteristics and its performance is higher compared to other frameworks. The proposed CNN is utilized for the classification of EEG signals into seizure, normal and preictal classes. The obtained results demonstrate high performance and good learning convergence of the proposed CNN framework.

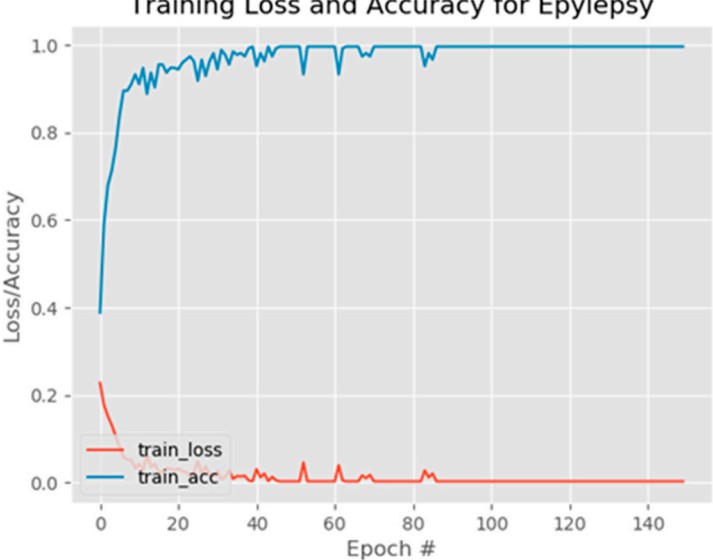

**Figure 4.** Simulation results using cross-validation.

**Table 4.** Simulation results using 10-fold cross-validation.

|  | Loss Function | Sensitivity (%) | Specificity (%) | Accuracy (%) |
|---|---|---|---|---|
| Train | $5.4302 \times 10^{-11}$ | 100 | 100 | 100 |
| Validation | 0.0177 | 96.30 | 98.77 | 97.53 |
| Test | 0.01771 | 97.67 | 98.83 | 98.67 |

In the next simulation, the presented CNN model was applied to the CHB-MIT scalp EEG dataset created by the investigators of Children's Hospital Boston (CHB) and the Massachusetts Institute of Technology (MIT) for the detection of seizure offset and onset in a long recording time frame. The EEG dataset of CHB-MIT scalp was recorded at the Children's Hospital at Boston. The dataset includes records of 23 pediatric patients with 844 h of continuous scalp EEG recording and 163 seizures. This dataset is comparatively large. By employing 256 samples/s, the EEG signals were digitized. The EEG dataset utilized was composed of 22 EEG channel signals from 18 patients with the same bipolar montages. The EEG dataset utilized in this paper is available for download from the PhysioNet website http://physionet.org/physiobank/database/chbmit/.

Most of the encephalogram recordings were contaminated by noise from the power line at 60 Hz. Because of the lack of availability of data in the database and distortions in some of the data, we used 7 patients with sufficient data for simulation in the paper. The CNN with the double convolutional layers was used for simulation purpose. Here, three double-convolutional layers are applied for extraction of features, and classification is performed using the fully connected layer.

During the simulation, training dataset size and system specifications were taken into account. Note that an Adam optimizer learning algorithm was utilized for training. The CNN includes four convolutional layers. A fully connected network with three layers was applied for classification purposes.

The CNN was trained using 150 epochs; 90% of the dataset was utilized for training purposes, and 10% for testing. At every iteration of each epoch, 70% of the training dataset was utilized for training, 30% for validation of the CNN. Out of 23 channels, the EEG signals of channels 15 and 23 were the same.

**Table 5.** Comparative results of different models.

| Author (Year) | Method | Performances |
|---|---|---|
| Ghosh-Dastidar et al. (2017) [54] | Spiking neural network | ACC: 92.5% |
| Ghosh-Dastidar et al. (2009) [55] | Levenberg–Marquardt backpropagation neural network | ACC: 96.7% |
| Ghosh-Dastidar et al. (2007) [12] | Principal component analysis enhanced cosine radial basis function neural network | ACC: 99.3% |
| Chua et al. (2010) [56] | Gaussian mixture model | ACC: 93.1%; SEN: 89.7% SPEC: 94.8% |
| Faust et al. (2010) [57] | SVM | ACC: 93.3%; SEN: 98.3% SPEC: 96.7% |
| Acharya et al. (2011) [58] | SVM-Discrete wavelet transform | ACC: 96.3%; SEN: 100% SPEC: 97.9% |
| Guo et al. (2011) [59] | Genetic Programming-KNN | ACC: 93.5% |
| Acharya et al. (2011) [16] | SVM | ACC: 95.6%; SEN: 98.9% SPEC: 97.8% |
| Acharya et al. (2012) [60] | Fuzzy Sugeno (Wavelet packet decomposition) | ACC: 96.7%; SEN: 95% SPEC: 99% |
| Martis et al. (2012) [61] | C4.5 decision tree | ACC: 95.3%; SEN: 98% SPEC: 97% |
| Bhattacharyya et al. (2017) [62] | Random forest | ACC: 99.4%; SEN: 97.9% SPEC: 99.5% |
| Bhattavharyya et al. (2017) [63] | SVM | ACC: 98.6% |
| Sharma et al. (2017) [64] | LS-SVM | SEN 100% ACC: 99%; SEN: 98% |
| Sharma et al. (2018) [47] | Orthogonal wavelet filters bank | SPEC: 99% |
| Acharya et al. (2018) [24] | Deep CNN | ACC: 88.7%; SEN: 95%; SPEC: 90% |
| Al-Sharhan et al. (2019) [43] | Genetic algorithm | ACC: 98.01%; SEN: 94.99%; SPEC: 98.65% |
| Gupta et al. (2019) [44] | FBSE + WMRPE + Regression | ACC: 98.6% |
| Vipani et al. (2017) [46] | Hilbert transform + Learning Vector Quantization | ACC: 89.31% |
| Ullah et al. (2018) [65] | P-1D-CNN | ACC: 99.6% |
| Thara et al. (2019) [66] | DNN | ACC: 97.21%; SEN: 98.59%; SPEC: 91.47% |
| Hassan et al. (2020) [67] | Complete ensemble empirical mode decomposition | ACC: 98.67%; SEN: 98.67%; SPE: 98.72% |
| Akyol (2020) [68] | SEA | ACC: 97.17%; SEN:93.11%; SPE: 98.18% |
| The current work | Deep CNN (10-fold cross-validation) | ACC: 98.67%; SEN: 97.67%; SPEC: 98.83% |

Abbreviation: ACC: accuracy; SEN: sensitivity; SPEC: specificity; KNN: *k*-nearest neighbors algorithm; LS-SVM: least-squares support vector machine; FBSE: Fourier–Bessel series expansion; WMRPE: weighted multiscale Renyi permutation entropy; P-1D-CNN: pyramidal one-dimensional convolutional neural network; DNN: deep neural network; SEA: stacking ensemble approach.

During the simulation, one of the problems that emerged during learning was over-fitting. Over-fitting is described as the situation where the model successfully learns the training dataset

(i.e., performs well on the training dataset) but does not perform well on a new or unknown dataset. Over-fitting happens when a model learns noise and detail in the training set to the extent that it negatively impacts the model's performance on new data. During the simulation, we applied some operations to solve the over-fitting problem, namely weight regularization, random shuffling of data and dropout. We performed weight regularization to improve the learning of the CNN. This allowed us to reduce over-fitting leading to faster optimization of the CNN model. We also shuffled the data before splitting it into the training and testing sets. In that way, the classes were equally distributed over the training and testing sets. Next, we applied the dropout operation. A deep neural network needs to learn a huge number of parameters, which in the case of a small dataset is likely to cause over-fitting. This issue is solved by designing dropout technology to prevent feature detectors from coadapting. The crucial concept of dropout is to drop units randomly from the neural network during training, with a predefined probability (along with their connections). Dropout methodology greatly eliminates over-fitting and provides major advantages over other forms of regularization. We introduced the dropout layer in the proposed model after the last ReLU activation function.

The constructed model integrates feature extraction and classification modules, which simplifies the structure of the epilepsy identification model. By including more convolutional layers or using other non-linear functions, we can increase the performance characteristics and also the accuracy of the model. However, this will lead to complication of the structure and learning process of the model. Consequently, decreasing the number of layers leads to a decrease in the accuracy rate of CNN. The results of the simulations are given in Table 6.

**Table 6.** Simulation results.

| Patient | No of Seizures | TP | FP | FN | SEN | FPR |
|---------|----------------|---------|-------|--------|-------|--------|
| Pat1 | 7 | 47.4286 | 0 | 3 | 94 | 0.0 |
| Pat2 | 3 | 17.33 | 38.33 | 33.33 | 33 | 4.381 |
| Pat5 | 5 | 40.2 | 3 | 11.6 | 76.85 | 1.0286 |
| Pat6 | 9 | 24.8889 | 4.33 | 26.667 | 46.92 | 1.4257 |
| Pat7 | 3 | 1 | 97.33 | 49 | 2 | 5.6974 |
| Pat9 | 4 | 15.25 | 12.25 | 35.25 | 29.65 | 0.98 |
| Pat23 | 5 | 46.54 | 0.8 | 4.4 | 91.4 | 0.274 |

Abbreviation: TP: true positive; FP: false positive; FN: false negative; SEN: sensitivity; FPR: false positive rate.

The goal of seizure detection is to segment the brain's electrical activity in real-time into seizure and non-seizure periods. This is implemented through the classification of extracted spectral and spatial features of the ECG signals. Rhythmic activity associated with the seizure is composed of multiple frequency components. It is necessary to consider multiple spectral components in order to improve the accuracy of seizure detection.

Epileptic patients have substantial EEG-related variation in non-seizure and seizure states. This affirms a steep trade-off between specificity and sensitivity of the model. We estimated the performance of the CNN model by using TP (true positive), FP (false positive), FN (false negative), sensitivity and false positive rate. Sensitivity refers to the percentage of identified test seizures. TP refers to the number of abnormal EEG recordings properly classified as being abnormal. True negative (TN) refers to the number of normal EEG cases properly classified as normal. FP refers to the number of normal EEG instances that are considered to be abnormal, while FN refers to the number of abnormal EEG recordings that are wrongly predicted as being normal. From Table 6, the most efficient performance metric of our model is sensitivity. This refers to the percentage of the correctly identified test seizures. If an alarm is triggered between its onset and its end, a seizure (true positive) is seen as correctly diagnosed. The rate of false alarm (false positive) refers to the average number of times,

per 24 h, the model erroneously predicts the onset of a seizure. Alarms that occur outside the interval between the beginning of a seizure and the end of the same seizure are considered false alarms.

The seizures differed from one another in the training dataset. The lack of sufficiently comparable cases of onset events from seizures in the training set led to the fairly low identification for each given test seizure. However, not all instances involving several distinct forms of seizures produce bad results. For instance, patient 6, for whom the clinical test identified a few types of seizure due to inadequate clinical symptoms, was increasingly achieving good results. This can theoretically be explained by the fact that the patient's data contained 9 seizures, and consequently, for most test seizures in this data, the model included more cases of large-type seizures and significant differences in the signal patterns between them was observed. The biggest estimate of the false alarm rate was obtained from patient 7. Some false alarms were the result of seizure onset activity mimics which many had morphological appearances like the those of ictal indications seen in various seizures. Furthermore, the lack of previous cases of comparable events in the interictal periods of the training set can clarify the huge number of false alarms for this record.

## 4. Conclusions

The design of a deep-learning structure based on CNNs for the detection of epilepsy using EEG signals has been performed. A four-level structure was used for training of the CNN to detect epilepsy. Training of CNN was implemented using a cross-validation technique. The design of the classification system was implemented using Bonn University data sets. As a result of the simulation, the average value of the accuracy rate for the test achieved was 98.67%, sensitivity was 97.67%, and specificity was 98.83%. One of the basic advantages of the model is its simple structure that combines the extraction of feature and classification stages in the body of the deep learning structure. The results obtained can be improved by increasing the number of convolutional layers, which in turn leads to complication of the deep structure. Future research is focusing on using a combination of CNNs and other feature extraction techniques to develop an identification system for the CHB-MIT dataset.

**Author Contributions:** Conceptualization: R.A.; investigation: M.A., B.S., J.B.I. and A.I.; methodology: R.A., M.A. and B.S.; software: M.A. and J.B.I.; supervision: R.A.; validation: M.A., B.S. and A.I.; resources: M.A., J.B.I. and A.I., Writing—Original draft: R.A., J.B.I. and M.A.; Writing—Review and editing: R.A. and J.B.I. All authors have read and agreed to the published version of the manuscript.

**Funding:** This research received no external funding.

**Conflicts of Interest:** The authors declare that there are no conflicts of interest regarding the publication of this paper.

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
