# Peer review of "Identification of Epileptic EEG Signals Using Convolutional Neural Networks"

_applsci, doi:10.3390/app10124089_

Round 1

Reviewer 1 Report

  • More discussion of the results needed. Whereas for the training and validations dataset the application works nicely, this is not the case for the second dataset. As usual the results show lower accuracy when dealing with different data, but, please, comment your results in more detail.
  • Seizure detection is highly sensitive to patient's conditions. No discussion is made related to this issue. Table 6 shows this.
  • Table 3 is confusing. What is the structure of the selected SVM and NN?

Author Response

Responses to the Comments

We would like to express our sincere thanks to the reviewers for their detailed analysis of the paper. We have addressed their comments in the paper, the changes made are highlighted in blue. 

Reviewer 1

Comments and Suggestions for Authors

  • More discussion of the results needed. Whereas for the training and validations dataset the application works nicely, this is not the case for the second dataset. As usual the results show lower accuracy when dealing with different data, but, please, comment your results in more detail.

Thank You for the comment. We extended the review and discussion sections. We added new references related to epileptic seizure detection published in recent years. The discussion section is extended. We also added future research to the conclusions. We have added the necessary explanation and discussion in the simulation section. We have downloaded the second data set from the web page https://physionet.org/content/chbmit/1.0.0/.  Because of data corruption, we utilized only the essential part of data sets. Future research is focusing on the modelling of the full data sets. More details have been added in section 3.

  1. Al-Sharhan S., Bimba A.. Adaptive multi-parent crossover GA for feature optimization in epileptic seizure identification . Applied Soft Computing. Applied Soft Computing Journal 2019; 75; 575–587.
  2. Gupta V., Pachori R.B.. Epileptic seizure identification using entropy of FBSE based EEG rhythms. Biomedical Signal Processing and Control 2019; 53; 1-11; 101569.
  3. George T.S., Subathra M.S.P., Sairamya N.J., Susmitha L., Premkumar J.M. Classification of epileptic EEG signals using PSO based artificial neural network and tunable-Q wavelet transform. Biocybernetics and Biomedical Engineering 2020; 40(2); 709-728.
  4. Vipani R., Hore S., Basu S., Basak S., Dutta S. Identification of epileptic seizures using Hilbert transform and learning vector quantization based classifier, in: IEEE Calcutta Conference (CALCON), 2017; 90–94.
  5. Sharma M., Bhurane A.A., Acharya U.R., MMSFL-OWFB: A novel class of orthogonal wavelet filters for epileptic seizure detection, Knowl.-Based Syst. 2018; 160; 265–277.

  • Seizure detection is highly sensitive to patient's conditions. No discussion is made related to this issue. Table 6 shows this.

Thank You for your valuable comment. Recordings were collected from 22 subjects (5 males, ages 3–22; and 17 females, ages 1.5–19). (Case CHB21 was obtained 1.5 years after case chb01, from the same female subject.) The goal of seizure detection is to segment the brain’s electrical activity in real-time into seizure and non-seizure periods. This is implemented through the classification of extracted spectral and spatial features of ECG signals. Rhythmic activity associated with the seizure is composed of multiple frequency components. It is necessary to consider multiple spectral components for detecting seizures with high accuracy.

Patients with epilepsy have considerable overlap in the EEG associated with seizure and non-seizure states. This forces to confront a steep trade-off between model sensitivity and specificity.   We estimate the CNN model performance using TP, FP, FN, sensitivity and false-positive rate. Sensitivity refers to the percentage of test seizures identified. True Positive (TP) is the number of abnormal EEG recordings which are correctly classified as abnormal. True Negative (TN) refers to the number of normal EEG instances which are correctly classified as normal. False-positive (FP) refers to a number of normal EEG instances that are identified to be abnormal. And false negative (FN) refers to the number of abnormal EEG recordings which are wrongly predicted to be normal. From table 6, the most efficient performance metric of our model is sensitivity. This refers to the percentage of test seizures correctly detected. A seizure (true positive) is viewed as correctly identified if an alarm is raised between its onset and its end. The false alarm rate (false positive) refers to the average number of times, per 24hrs, that the model erroneously detects the onset of a seizure. Alarms that begin outside intervals between seizure onset and the end of the same seizure are referred to as false alarms.

Seizures in the training dataset are varied from each other. For each given test seizure, the lack of adequately comparable instance of onset activity from seizures in the training data contributed to the relatively poor detection. Although not all cases that include multiple distinct seizure types yield poor outcomes. For instance, progressively good outcomes were achieved for patient 6, for whom the clinical test listed a few seizure types due to insufficient of clinical symptoms. This is potentially clarified by the fact that the data of the patient includes 9 seizures, and subsequently, for most test seizures in this dataset, the model has incorporated more instances of seizures of the wide type, and observed notable variation in the signal patterns among them. The biggest false alarm rate estimate was gotten from patient 7. Some false alarms were as a result of mimics of seizure onset activity where many had a morphological appearance like that of ictal indications seen in different seizures. Also, lack of prior instances of comparable activity in the interictal periods in the training data may clarify the huge number of false alarms for this record.

  • Table 3 is confusing. What is the structure of the selected SVM and NN?

Thank You for the comment. The structures of the selected models are provided. In the paper we applied SVM model for comparison purpose. We used linear SVM with linear kernel function. We also applied three layer NN structure with 18 hidden neurons and Gaussian activation function.

Reviewer 2 Report

The article presents the idea of a CNN-based epileptic seizure predicator. Advantages of the article: 1) The CNN for classification of EEG data is given. 2) The CNN is trained on the basis of data from an open source. 3) Results are compared with other solutions known from the literature. Disadvantages of the article: 1) A comparison with old methods is made - only two solutions younger than 5 years are taken into account. 2) It is not clear whether all the classifiers compared were trained, evaluated and tested using the same data and methodology. 3) There is no evidence that over-fit to training data has been eliminated and that differences between models are statistically significant. 4) It is extremely difficult for the reader to repeat the author simulations without the code used by the author. 5) Table 4 probably contains mixed data. Some are percentages, some are fractions.

Author Response

Responses to the Comments

We would like to express our sincere thanks to the reviewers for their detailed analysis of the paper. We have addressed their comments in the paper, the changes made are highlighted in blue. 

Reviewer 2

Comments and Suggestions for Authors

The article presents the idea of a CNN-based epileptic seizure predicator. Advantages of the article: 1) The CNN for classification of EEG data is given. 2) The CNN is trained on the basis of data from an open source. 3) Results are compared with other solutions known from the literature.

Disadvantages of the article: 1) A comparison with old methods is made - only two solutions younger than 5 years are taken into account.

Thank You for the comment. According to this comment we have done corrections in the paper. We added new papers published in the last two years. We include the results of these papers to Table 5.

1.Salah Al-Sharhan, Andrew Bimba.  Adaptive multi-parent crossover GA for feature optimization in epileptic seizure identification. Applied Soft Computing. Applied Soft Computing Journal 75 (2019) 575–587

2. Vipin Gupta, Ram Bilas Pachori. Epileptic seizure identification using entropy of FBSE based EEG rhythms. Biomedical Signal Processing and Control 53 (2019) 101569

3. R. Vipani, S. Hore, S. Basu, S. Basak, S. Dutta, Identification of epileptic seizures using Hilbert transform and learning vector quantization based classifier, in: Calcutta Conference (CALCON), 2017 IEEE, IEEE, 2017, pp. 90–94.

4. M. Sharma, A.A. Bhurane, U.R. Acharya, MMSFL-OWFB: A novel class of orthogonal wavelet filters for epileptic seizure detection, Knowl.-Based Syst. 160 (2018) 265–277.

5. S. Thomas George, M.S.P. Subathra, N.J. Sairamya, L. Susmitha,M. Joel Premkumar

Classification of epileptic EEG signals using PSO based artificial neural network and tunable-Q wavelet transform. Biomedical Signal Processing and Control 53 (2019) 101569

2) It is not clear whether all the classifiers compared were trained, evaluated and tested using the same data and methodology.

Thank You for the comment. The comparisons of classifiers are done using the same data sets. (Bonn data sets). In the simulation, we have used the partitions of data that was presented by authors [18] of Bonn data sets. The models provided in Table 5 utilized the same data sets (Bonn data set). The comparison is provided using the same initial conditions.  We added some explanations in order to answer the question in the comment. 

3) There is no evidence that over-fit to training data has been eliminated and that differences between models are statistically significant.

Thank You for the comment. Overfitting refers to a model that learns the training dataset well, performs well on training dataset but does not perform well on new, unseen data. Overfitting happens when a model learns the detail and noise in the training data to the extent that it negatively impacts the performance of the model on new data. During the simulation, we applied some operations to solve the over-fitting problem. These are weight regularization, random shuffling of data and dropout. We did weight regularization for improving the learning of CNN. This allows us to reduce over-fitting and leading to faster optimization of CNN model. We also shuffled the data before splitting them into train and test set. In that way, the classes are equally distributed over the train and test sets. Next, we apply the dropout operation. A deep neural network needs to learn a large number of parameters, which is likely to cause over-fitting in the case of a small dataset. We addressed this issue by developing dropout technology to prevent the coadaptation of feature detectors. The critical idea of dropout is to randomly drop units with a predefined probability (along with their connections) from the neural network during training. Dropout technique significantly reduces overfitting and gives significant improvements over other regularization methods. In the proposed model, we added the dropout layer after the last ReLu activation function.

4) It is extremely difficult for the reader to repeat the author simulations without the code used by the author.

Thank You for the comment. After publication, we will provide the open code for the interested researchers. Some explanation about simulation is provided. Training of the proposed model needs the weight parameters to be learned from the EEG data. For learning these parameters, we employed RMSprop algorithm. The modelling has been done in Keras, a powerful deep learning library, which runs on top of TensorFlow. The batch size of 100 is chosen in this work, which is used for each training update. To compare the performance measure, we trained all the models that are present in this work with 150 epochs.

5) Table 4 probably contains mixed data. Some are percentages, some are fractions.

Thank You for the comment. According to this comment, we have done the necessary corrections in the paper. All results are presented using a percentage.

Top of Form

Reviewer 3 Report

This paper demonstrated the Epileptic EEG classification using CNN. The accuracies were high but the writing requires much improvement. A number of details of their method are missing and the comparison of the results is not good enough. I suggest the manuscript should be re-organized.

Some minor comments are given as follows:

-In title, S should be capitalized:'S'ignal
-Give the full name of EEG at the first appearance in the Introduction.
-Give the full name of WT (much more to be added except of this one).
-CNN "iis" a multilayer neural network
-The tense of the waiting needs to be thoroughly reconsidered.
-Please give details of your methods. For example, One dimensional double CNN? What does that mean and how it works?
-Number of convolutional layers should be provided instead of just saying "is set by the programmer"
-How did the inout size (4096) come from? How many channels were used?
-Figure 3 is not necessary.
-The sensitivity and specificity can be calculated by using SVM with probability.

Author Response

Responses to the Comments

We would like to express our sincere thanks to the reviewers for their detailed analysis of the paper. We have addressed their comments in the paper, the changes made are highlighted in blue. 

Reviewer3

Comments and Suggestions for Authors

This paper demonstrated the Epileptic EEG classification using CNN. The accuracies were high but the writing requires much improvement. A number of details of their method are missing and the comparison of the results is not good enough. I suggest the manuscript should be re-organized.

Thorough proofreading and possible corrections have been done.

Some minor comments are given as follows:

-In the title, S should be capitalized:'S'ignal

The correction has been done

-Give the full name of EEG at the first appearance in the Introduction.

The correction has been done

-Give the full name of WT (much more to be added except of this one).

The correction has been done

-CNN "iis" a multilayer neural network

Thank You for the comments. All the corrections have been done

-The tense of the waiting needs to be thoroughly reconsidered.

We tried our best to do corrections in a short time.

-Please give details of your methods. For example, One dimensional double CNN? What does that mean and how it works?

Thank You for the comment. Because each convolution layer includes two layers, we called the network double convolutional neural networks. Here instead of a set of convolutional filters that are independently learned, we are using a group of filters. Here the network uses several two-step convolutional procedure. The presented CNN has advantage on ordinary (having one convolutional layer) CNN. The obtained results demonstrate the advantage of double convolutional CNN as shown in the simulation section.

Since our input data is a 1-dimensional brain signal, our CNN model is designed to accept 1-dimensional data which is 4096 length according to the dataset specifications. After getting input data, the first convolutional operation is applied to the data and then the second convolutional operation is applied to result of the first convolutional operation. Afterwards, the pooling is applied to the outputs of the convolutional layer. Pooling decreases the dimensions of the data. Increasing the number of convolutional layers allows us to obtain more deep features, but this increase computational time.   As a result, we would typically like to have multiple convolutional layers before a pool, so that we can build up better representations of the data without quickly losing all of your spatial information.

-Number of convolutional layers should be provided instead of just saying "is set by the programmer"

Thank You for the comment. The correction has been done. A number of simulations have been done in order to enable the CNN structure to provide good performance characteristic. As a result of the analysis, we explored four double-convolutional layers as demonstrated in Figure 1.

-How did the input size (4096) come from? How many channels were used?

Thank You for the comment. The BU dataset is obtained from five patients and each dataset comprises of 100 EEG signals. Each signal is recorded for 23.6 seconds. In simulations, EEG signals with the same 128 channel amplifier are recorded. The data is digitalized at 12 bits resolution and 173.6 samples per second. The bandwidth of the acquisition system is 0.5Hz to 85Hz. Each dataset has 4096 (23.5*173.6) sampling points for 23.5 seconds. The total number of EEG signals used in the simulation is 300, where 100 normal signals for (Set B), 100 preictal for (Set D) and 100 seizures for (Set E). We use these three classes of data in the simulations.

-Figure 3 is not necessary.

Thank You for the comment. The correction has been done. Fig.3 has been removed.

-The sensitivity and specificity can be calculated by using SVM with probability.

Thank You for the comment. The corrections have been done. After the simulation, we added sensitivity and specificity as provided in the table.

Round 2

Reviewer 3 Report

I would suggest the authors to carefully check their manuscript again before submission. Many flaws appeared in the manuscript and I was unable to go through it smoothly in the current form. Here I am listing some errors:

  1. Please avoid using contractions such as "won't" in academic writing, in page 2
  2. EEG does not stand for encephalogram.
  3. Use the abbreviation after you have defined it. For example, in page 2, "Using the encephalogram..." But you have already defined "encephalogram".
  4. Again, please check all the abbreviations are mentioned before used. For example, SVM, LMT, PSO...
  5. In page 3, insert a blank before [46]: classifier[46]
  6. In equation (1), the first l+1, I, j, k should not be in bold face since they represent single values.
  7. The authors mentioned that they added a dropout layer to their CNN model. I did not see it in Table 1.
  8. It is CHB-MIT, not CHI-MIT.

Author Response

Responses to the Comments

We would like to thank You for the analysis and review of the paper. We have addressed the comments in the paper, the changes made being in blue colour. 

I would suggest the authors to carefully check their manuscript again before submission. Many flaws appeared in the manuscript and I was unable to go through it smoothly in the current form.

Response: Thank You for the comment. In the revised version, we have checked the paper, and some grammar errors have corrected. We tried our best to improve the paper. We wish the current version can satisfy you.

Here I am listing some errors:

1. Please avoid using contractions such as "won't" in academic writing, in page

Response: Thank You for the comment. The correction has been done

2. EEG does not stand for encephalogram.

Response: Thank You for the comment. It is right. The functions of EEG and encephalogram are different. We did some changes in the text to reflect this idea.

3. Use the abbreviation after you have defined it. For example, in page 2, "Using the encephalogram..." But you have already defined "encephalogram".

Response: Thank you for the comment. It should be EEG signals. The correction has been done

4. Again, please check all the abbreviations are mentioned before used. For example, SVM, LMT, PSO...

Response: Thank you for the comment. The corrections have been done

5. In page 3, insert a blank before [46]: classifier[46]

Response: Thank you for the comment. The correction has been done

6. In equation (1), the first l+1, I, j, k should not be in bold face since they represent single values.

Response: Thank you for the comment. The correction has been done

7. The authors mentioned that they added a dropout layer to their CNN model. I did not see it in Table 1.

Response: Thank you for the comment. In the given research we used one dropout layer with the probability of 0.5. The dropout layer used is added to the Table 1.

8. It is CHB-MIT, not CHI-MIT.

Response: Thank you for the comment. The correction has been done.

Round 3

Reviewer 3 Report

Please carefully go through the manuscript before publishing.

I can still find some flaws in it:

1. In equation (5), the first L should be L(θ)

2. Again, in line 367: "We've" 

3. "CNN was trained using 150 epochs", which is mentioned three times in line 274, 269, and 258 as well as in 345 and 350. Besides, some information was also repeatedly given. 

4. In line 368, ReL"U"

5. Please make sure the decimal places were not missing for SVM and NN.

Author Response

Responses to the Comments

We would like to thank You for the useful comments. We checked the paper and have done English editing. In the revised version we have addressed the comments. We wish the current version can satisfy you.

I can still find some flaws in it:

In equation (5), the first L should be L(θ).  

Thank You for the comment. The correction has been done

  1. Again, in line 367: "We've" 

Thank You for the comment. In the revised version, we have done English editing.

  1. "CNN was trained using 150 epochs", which is mentioned three times in line 274, 269, and 258 as well as in 345 and 350. Besides, some information was also repeatedly given. 

Thank You for the comment. In the revised version the repeated sections were deleted. In the paper, we have given the results of three simulations. Two of them were related to Bonn dataset, one-  CHI MIT dataset. The parameters used in learning were mentioned for each simulation.

  1. In line 368, ReL"U"

Thank You for the comment. In the revised version, we have checked the paper, and some typos and grammatical errors have corrected.

  1. Please make sure the decimal places were not missing for SVM and NN.

We presented the results of the simulation. Because of the decimal places are very small we rounded them up.
